# Mitigating Sample Selection Bias with Robust Domain Adaption in Multimedia Recommendation

## ABSTRACT

Industrial multimedia recommendation systems extensively utilize cascade architectures to deliver personalized content for users, generally consisting of multiple stages like retrieval and ranking. However, retrieval models have long suffered from Sample Selection Bias (SSB) due to the distribution discrepancy between the exposed items used for model training and the candidates (almost unexposed) during inference, affecting recommendation performance. Traditional methods utilize retrieval candidates as augmented training data, indiscriminately treating unexposed data as negative samples, which leads to inaccuracies and noise. Some efforts rely on unbiased datasets, while they are costly to collect and insufficient for industrial models. In this paper, we propose a debiasing framework named DAMCAR, which introduces Domain Adaptation to mitigate SSB in Multimedia CAscade Recommendation systems. Firstly, we sample hard-to-distinguish samples from unexposed data to serve as the target domain, optimizing data quality and resource utilization. Secondly, adversarial domain adaptation is employed to generate pseudo-labels for each sample. To enhance robustness, we utilize Exponential Moving Average (EMA) to create a teacher model that supervises the generation of pseudo-labels via self-distillation. Finally, we obtain a retrieval model that maintains stable performance during inference through a hybrid training mechanism. We conduct offline experiments on two real-world datasets and deploy our approach in the retrieval model of a multimedia video recommendation system for online A/B testing. Comprehensive experimental results demonstrate the effectiveness of DAMCAR in practical applications.

## CCS CONCEPTS

• **Information systems** → **Recommender systems**.

## KEYWORDS

Multimedia recommendation, Debiasing, Cascade systems

## 1 INTRODUCTION

Multimedia recommendation is essential for many mobile Internet platforms, aiming to accurately present multi-modal items aligned with users' interests [36, 40, 53, 57, 58, 62, 64]. To support the increasing demand for online deployment, industrial scenarios have

**Unpublished working draft. Not for distribution.**

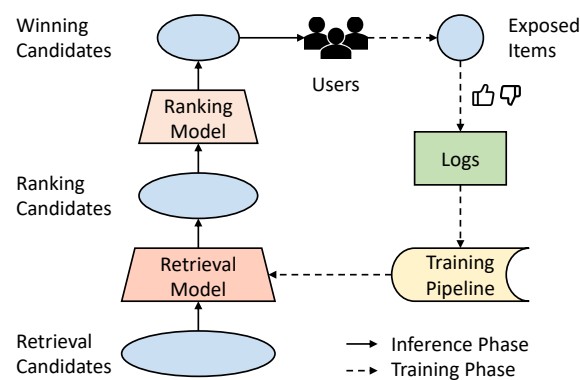

**Figure 1: An illustration of the inference and training phases in a typical multimedia cascade recommendation system.**

widely adopted cascade architectures [38, 48, 54, 56]. Such architectures utilize a funnel-shaped filtering strategy—from broad retrieval to precise ranking—to efficiently deliver personalized content for users. Figure 1 illustrates a typical multimedia cascade recommendation system comprising two stages: retrieval and ranking[1]. Simple models are employed in the retrieval stage to swiftly filter out irrelevant items from a large candidate pool [10, 28, 49]. The ranking stage then employs sophisticated models for precise ranking, selecting the top few as winning candidates [3, 44, 45, 65, 66]. Finally, those exposed items with user behaviors (e.g., finish playing or not) are recorded for continuous model training [7, 19].

However, in cascade architectures, retrieval models face the challenge of *Sample Selection Bias* (SSB) [24, 33, 36, 61]. As illustrated in Figure 1, this bias originates from a notable phenomenon: There exists a significant discrepancy between the distribution of exposed items used for model training and the distribution of retrieval candidates (almost unexposed) during inference. Such bias impedes the ability of retrieval models to comprehensively learn the distribution characteristics across the entire data space, affecting models' generalization performance and failing to reflect users' real interests [7, 24]. In essence, SSB violates the foundational training-inference consistency assumption [19, 48], like requiring a student to take an exam beyond the prescribed syllabus.

To mitigate SSB, traditional methods utilize retrieval candidates for data augmentation, indiscriminately treating unexposed data as negative samples [7, 24, 27]. Although reducing the distribution discrepancy, they overlook the potential positive samples within unexposed data. Such mislabeling introduces training noise, diminishing the quality of decision-making during inference.

Another line of research attempts to collect unbiased datasets through specific uniform strategies and then guide the models

---

[1]In fact, multimedia cascade recommendation systems also include a pre-ranking stage to bridge retrieval and ranking [31, 32]. However, our study primarily focuses on retrieval models, excluding the pre-ranking stage for simplicity.

trained on exposed data by knowledge transfer [33–35, 37, 63]. However, these solutions encounter prohibitive data collection and processing costs, making them impractical for real-world applications [36]. Furthermore, the obtained unbiased datasets may be insufficient for industrial models with billions of parameters [19].

Given that the primary cause of SSB is the distribution inconsistency between training and inference data, we draw on unsupervised domain adaptation techniques to tackle this issue [41, 60]. Unsupervised domain adaptation has demonstrated notable success in fields like image recognition [4, 26] and natural language processing [2, 5], by aligning distributions of the labeled source domain and the unlabeled target domain to ensure training-inference consistency. However, when applying it to resolve SSB, researchers face several key challenges: (i) Unsupervised domain adaptation requires unbiased datasets as target domains for model training, which is often difficult to achieve in practical applications [36, 56]. (ii) There is lacking effective mechanisms to guarantee the quality of the pseudo-labels generated for target domains, while they are critical to model performance [18, 51].

In this paper, we propose a novel debiasing framework named **DAMCAR**, which introduces **D**omain **A**daptation to mitigate SSB in **M**ultimedia **CA**scade **R**ecommendation systems. DAMCAR consists of three modules: (i) Target domain generation. We construct a directed weighted bipartite graph from user historical behaviors and employ a random walk algorithm to select hard-to-distinguish samples from unexposed data to serve as the target domain. This method not only obtains informative samples for model training but is also resource-friendly by focusing only on a subset of unexposed data rather than all of them. (ii) Robust pseudo-label generation. We design a label generator that introduces an adversarial network between the source domain (exposed data) and the target domain (sampled unexposed data). This encourages the model to learn feature representations that are indistinguishable across data distributions, thereby generating unbiased pseudo-labels for each sample. Additionally, to improve the model's robustness and the reliability of pseudo-labels, we use Exponential Moving Average (EMA) to create a teacher model that guides the generation of pseudo-labels via self-distillation. (iii) Hybrid training. We utilize a hybrid training mechanism that combines both pseudo-labels and the scores from the ranking model to obtain a retrieval model that maintains stable performance during inference. Compared with traditional solutions, DAMCAR provides customized samples and corresponding labels to the retrieval model at the end of each training phase without affecting its efficiency. To summarize, we make the following contributions in this paper:

- We propose DAMCAR, aimed at mitigating SSB by leveraging domain adaptation to bridge the gap between training and inference data distributions for retrieval models, which focuses on the generation of the target domain and pseudo-labels.
- To enhance robustness and reliability, we use EMA to create a teacher model that steers the generation of pseudo-labels for target domain samples via a self-distillation mechanism.
- We conduct offline experiments on two datasets and implement our approach in a multimedia video recommendation system for online A/B testing. Experimental results demonstrate the effectiveness of DAMCAR in real-world applications.

## 2 RELATED WORKS

### 2.1 Multimedia Cascade Recommendation

In industrial scenarios, the online deployment of multimedia recommendation systems necessitates balancing effectiveness and efficiency. While sophisticated models offer superior recommendation performance, they tend to introduce high latency [36, 43, 57]. Conversely, simple models with limited capacity can efficiently handle massive requests [28, 67]. Thus, a prevalent approach is to adopt funnel-shaped cascade architectures [17, 25, 38, 48, 54, 56]. Simple models are employed in the early retrieval stage to swiftly filter out irrelevant items from a large candidate pool [10, 28, 49], while sophisticated models are then utilized in the ranking stage for precise ranking [3, 44, 45, 65, 66]. Some studies explore cost-aware cascading systems that incorporate computational cost to determine model assignments for different stages [8, 54, 59]. Other solutions optimize cascade systems through gradient transfer, stage combination, feature sharing, and item aggregation [15–17, 25]. Despite these advancements, most existing multimedia cascade recommendation systems suffer from SSB due to the distribution discrepancy between training and inference data [48, 56], potentially impacting model performance. In this paper, we focus on mitigating SSB by providing customized samples and corresponding labels to retrieval models without compromising the training efficiency.

### 2.2 Debiasing in Cascade Systems

Many recent efforts are devoted to mitigating SSB for improved recommendation performance [19, 27, 33, 48, 63]. (i) Some approaches focus on the augmentation of training data. They either employ all retrieval candidates, treating unexposed data as negative samples [7, 24] or utilize random negative sampling and hard negative sampling to select negative samples [11, 27, 29, 48]. These methods reduce the distribution discrepancy, but overlook the potential positive samples within unexposed data, introducing training noise. (ii) Another line of research is to generate unbiased datasets by employing specific uniform strategies for content delivery and collecting user feedback [33, 37]. The unbiased datasets are used to train imputation models for unexposed data [1, 6, 13, 63] or guide the models trained on exposed data by knowledge transfer [33–35, 37]. However, these approaches incur high costs in unbiased dataset collection and processing that may hurt the user experience, rendering them impractical for real-world applications [19, 36]. (iii) Unsupervised domain adaptation is a potential research direction for mitigating SSB [4, 26], which trains a model that performs well on the unlabeled target domain by using labeled samples on the source domain. For example, in [56], a scoring model is trained on exposed data (source domain) and generates pseudo-labels for an unbiased dataset (target domain). The retrieval model is trained on the target domain, ensuring the training-inference consistency. However, it requires an unbiased dataset as the target domain for model training, which is often difficult to achieve in practical applications [19, 36]. Moreover, there is currently lacking effective mechanisms to ensure the quality of the pseudo-labels generated for target domains [18, 51]. Building upon existing research, we leverage unexposed data to generate a target domain with informative samples and develop a robust pseudo-label generation mechanism, aiming to mitigate SSB from two aspects of data and labels.

**Figure 2: The architecture of DAMCAR, which consists of three modules: (i) Target domain generation (left). We generate a high-quality unexposed sample set $\mathcal{S}_{sam}$ from $\mathcal{S} \setminus \mathcal{S}_{exp}$ to serve as the target domain, by constructing a directed weighted bipartite graph and employing a random walk algorithm. (ii) Robust pseudo-label generation (middle). We generate pseudo-labels for each sample through adversarial domain adaptation and then use EMA to create a teacher model that guides the generation via self-distillation for enhanced robustness. (iii) Hybrid training (right). We train the retrieval model Ret($\cdot$) through a hybrid mechanism that combines modified pseudo-labels with the scores obtained from the ranking model Rank($\cdot$).**

## 3 PRELIMINARIES

In this section, we provide a formal definition of the problem along with the necessary notations. For a multimedia recommendation system, it includes a user set $\mathcal{U} = \{u_1, \ldots, u_N\}$ and an item set $\mathcal{I} = \{i_1, \ldots, i_M\}$, where $N$ represents the number of users and $M$ represents the number of items. We define $\mathcal{S} = \mathcal{U} \times \mathcal{I}$ as the entire sample set, encompassing all user-item pairs. In the cascade architecture, the retrieval model Ret($\cdot$) initially scores the entire sample set $\mathcal{S}$ and selects a subset with high scores. The subset is then forwarded to the ranking model Rank($\cdot$) for further processing, ultimately delivering recommendation results for users. After collecting user interactions, an exposed sample set $\mathcal{S}_{exp} = \{(u, i, y, e = 1) \mid u \in \mathcal{U}, i \in \mathcal{I}, y \in \{0, 1\}\}$ is produced:

- $u$: the features of a user, which include personal attributes and historical behaviors.
- $i$: the features of an item, which include item attributes like ID, texts, images, and other multi-modal information.
- $y$: the ground-truth label of an observed user behavior, e.g., finish playing ($y = 1$) or not ($y = 0$).
- $e$: the exposure label, where $e = 1$ denotes item $i$ has been exposed to user $u$ with corresponding behavior $y$.

The ranking model Rank($\cdot$) is trained on the exposed sample set $\mathcal{S}_{exp}$ [19, 48, 56], which typically employs a sophisticated structure to deeply interact with user interests and item features for precise ranking, e.g., Deep Factorization-Machine (DeepFM) [22] and Deep & Cross Network (DCN) [55]. On the other hand, the retrieval model Ret($\cdot$) adopts a simple dual-tower structure to match

user features with item features, focusing on efficiently selecting potentially user-interested items from a large candidate pool, e.g., Deep Structured Semantic Model (DSSM) [28]. However, in the cascade architecture, the retrieval model Ret($\cdot$) suffers from SSB, as highlighted in previous studies [24, 33, 36, 61]. This issue arises from a notable phenomenon: Training Ret($\cdot$) on a limited labeled subset $\mathcal{S}_{exp}$ comprising only exposed items while requiring it to score the entire sample set $\mathcal{S}$ with mostly unexposed items during inference. SSB causes the training-inference inconsistency, which may impact the model's recommendation performance.

## 4 METHOD

In this paper, we propose a debiasing framework named **DAMCAR** to mitigate SSB in multimedia cascade recommendation systems, which consists of three modules: (i) Target domain generation (§4.1), (ii) Robust pseudo-label generation (§4.2), and (iii) Hybrid training (§4.3). Figure 2 illustrates the architecture of DAMCAR.

### 4.1 Target Domain Generation

To optimize data quality and resource utilization, we collect an unexposed sample set $\mathcal{S}_{sam} = \{(u, i, e = 0) \mid u \in \mathcal{U}, i \in \mathcal{I}\}$ from $\mathcal{S} \setminus \mathcal{S}_{exp}$ to serve as the target domain, rather than utilizing all of them. In contrast to existing sampling methods [11, 27, 29, 48], our focus is on selecting hard-to-distinguish samples that are difficult for the model to judge, containing more information in enhancing training. These unlabeled samples are equipped with pseudo-labels in §4.2, instead of being treated indiscriminately as negative.

Specifically, we leverage graph connectivity to identify the degree to which unexposed samples are close to the model's decision boundary. This graph-based perspective compensates for the insufficient exposure at the individual user-item pair level by exploiting the inherent multi-hop relationships in the network structure. It enables the computation of connectivity between users and items, even without direct links, which is particularly crucial for samples that are inherently hard-to-distinguish.

We start by constructing a directed weighted bipartite graph with the exposed sample set $\mathcal{S}_{exp}$, where nodes represent users or items and edges represent interactions. The weight $v_{u,i}$ denotes the degree of engagement between user $u$ and item $i$, such as watch time. We define $\mathcal{P}(u \rightarrow i)$ as the transition probability from user $u$ to item $i$, which is calculated as the ratio of the weight $v_{u,i}$ to the sum of weights for all items that user $u$ has interacted with:

$$\mathcal{P}(u \rightarrow i) = \frac{v_{u,i}}{\sum_{i' \in \mathcal{I}(u)} v_{u,i'}}, \tag{1}$$

where $\mathcal{I}(u)$ is the set of item nodes directly connected with user $u$.

Similarly, the transition probability from item $i$ to user $u$, denoted as $\mathcal{P}(i \rightarrow u)$, is determined by comparing the weights $v_{u,i}$ with the sum of weights from all users who have engaged with item $i$:

$$\mathcal{P}(i \rightarrow u) = \frac{v_{u,i}}{\sum_{u' \in \mathcal{U}(i)} v_{u',i}}, \tag{2}$$

where $\mathcal{U}(i)$ is the set of user nodes directly connected with item $i$.

Now, we apply a random walk algorithm with restart mechanisms to obtain hard-to-distinguish samples for each user from $\mathcal{S} \setminus \mathcal{S}_{exp}$. We center on user $u$ with a radius of $R$ and randomly visit nodes along the path of $u \rightarrow i \rightarrow u$, according to the transition probability $\mathcal{P}$. When the next hop exceeds the radius $R$, the process returns to the starting node $u$ for resampling, which continues until $H$ visits are completed. Let $\mathcal{T}(u, i)$ represent the number of times each item $i$ is visited. We denote the set of visited exposed items as $\mathcal{I}_u = \{i \mid (u, i, y, e = 1), \mathcal{T}(u, i) \geq 1\}$ and the set of visited unexposed items as $\mathcal{I}_u^* = \{i^* \mid (u, i^*, e = 0), \mathcal{T}(u, i^*) \geq 1\}$. Then, we define hard-to-distinguish samples as follows:

$$\{(u, i^*, e = 0) \mid i^* \in \mathcal{I}_u^*, \mathcal{T}(u, i^*) \geq \frac{1}{|\mathcal{I}_u|} \sum_{i \in \mathcal{I}_u} \mathcal{T}(u, i)\}. \tag{3}$$

That is, when the number of visits $\mathcal{T}(u, i^*)$ for an unexposed item $i^* \in \mathcal{I}_u^*$ is greater than or equal to the average number of visits for exposed items $\{i \mid i \in \mathcal{I}_u\}$, the sample $(u, i^*, e = 0)$ will be added to the unexposed sample set $\mathcal{S}_{sam}$. $|\mathcal{I}_u|$ represents the number of visited exposed items for user $u$.

## 4.2 Robust Pseudo-Label Generation

Here, we define exposed samples as the source domain $\mathcal{S}_{exp} = \{(u^s, i^s, y^s, e = 1)\}$ and sampled unexposed samples as the target domain $\mathcal{S}_{sam} = \{(u^t, i^t, e = 0)\}$ for greater clarity. We introduce adversarial domain adaptation to learn exposure-independent pseudo-labels for retrieval model training.

Initially, we employ a unified encoder $\mathcal{E}(\cdot)$ to extract features from samples in both the source and target domains, yielding their deep representations $\mathbf{z}^s$ and $\mathbf{z}^t$, respectively:

$$\mathbf{z}^s, \mathbf{z}^t = \mathcal{E}(u^s, i^s), \mathcal{E}(u^t, i^t). \tag{4}$$

Then, we feed the deep representations $\mathbf{z}^s$ and $\mathbf{z}^t$ into the label generator $\Phi(\cdot)$ to generate the pseudo-labels $\hat{y}^s$ for source domain samples and the pseudo-labels $\hat{y}^t$ for target domain samples:

$$\hat{y}^s, \hat{y}^t = \Phi(g(\mathbf{z}^s)), \Phi(\mathbf{z}^t), \tag{5}$$

where $g(\cdot)$ is the linear transformation. In this stage, we optimize the model's generation accuracy by minimizing the binary cross-entropy loss [12] on source domain samples:

$$\mathcal{L}_{ce} = -\frac{1}{|\mathcal{S}_{exp}|} \sum_{\mathcal{S}_{exp}} y^s \log \hat{y}^s + (1 - y^s) \log(1 - \hat{y}^s), \tag{6}$$

where $|\mathcal{S}_{exp}|$ is the number of source domain samples.

To mitigate SSB, we aim to align the distributions of the source and target domains, ensuring that the learned deep representations of samples are exposure-independent. For this purpose, we incorporate a domain classifier $\Psi(\cdot)$ that predicts whether a sample belongs to the source or target domain and is trained in an adversarial manner. Specifically, we desire the domain classifier $\Psi(\cdot)$ to make accurate predictions (minimize the training loss), and then through Gradient Reversal Layer (GRL) [18], we compel the encoder $\mathcal{E}(\cdot)$ to best fool $\Psi(\cdot)$, i.e., making $\mathbf{z}^s$ and $\mathbf{z}^t$ indistinguishable across data distributions as much as possible (maximize the training loss). The process can be formulated as follows:

$$\hat{d}^s, \hat{d}^t = \Psi(g(\mathbf{z}^s)), \Psi(\mathbf{z}^t), \tag{7}$$

$$\mathcal{L}_{dom} = -\frac{1}{|\mathcal{S}_{exp}|} \sum_{\mathcal{S}_{exp}} \log \hat{d}^s - \frac{1}{|\mathcal{S}_{sam}|} \sum_{\mathcal{S}_{sam}} \log(1 - \hat{d}^t), \tag{8}$$

where $|\mathcal{S}_{sam}|$ is the number of target domain samples and $\mathcal{L}_{dom}$ is the training loss of domain classifier $\Psi(\cdot)$. In this way, we tightly align the representations of source domain samples and target domain samples, generating unbiased pseudo-labels for each sample.

However, this unsupervised domain adaptation approach cannot guarantee the quality of the pseudo-labels generated for the target domain. To address this issue, we incorporate a self-distillation mechanism to supervise the learning of pseudo-labels, thereby enhancing model robustness and label quality. Inspired by ensemble learning techniques [9, 30], which posit that a combination of historical models yields greater performance than a single model, we treat each batch's update as the student model, and an ensemble of student models across multiple batches forms the teacher model. For updates of the teacher model's parameters, we employ Exponential Moving Average (EMA) to control the influence of the newly updated student model on it as follows:

$$\Theta_{tea}^{(t)} = \alpha * \Theta_{tea}^{(t-1)} + (1 - \alpha) * \Theta_{stu}^{(t)}, \tag{9}$$

where parameters of the teacher model in batch $t$ are updated from the corresponding parameters of the student model, with a smoothing parameter $\alpha \in (0, 1)$. We can elevate the student model to a teacher model without altering the original model structure. Subsequently, we use the outputs $y_{tea}^t$ generated by the teacher model to supervise the learning of pseudo-labels for target domain samples, with the training loss defined as follows:

$$\mathcal{L}_{sup} = -\frac{1}{|\mathcal{S}_{sam}|} \sum_{\mathcal{S}_{sam}} y_{tea}^t \log \hat{y}^t + (1 - y_{tea}^t) \log(1 - \hat{y}^t). \tag{10}$$

Overall, to generate robust unbiased pseudo-labels, we need to minimize the two losses for label classification ($\mathcal{L}_{ce}$ and $\mathcal{L}_{sup}$)

---

**Algorithm 1** Hybrid Training Process

---

    **Input:** Training sample set $\mathcal{S}_T = \mathcal{S}_{exp} \cup \mathcal{S}_{sam}$, Ranking model
    Rank$(\cdot)$, Encoder $\mathcal{E}(\cdot)$
    **Output:** Retrieval model Ret$(\cdot)$
1: **for** each epoch **do**
2:    **for** each batch in $\mathcal{S}_T$ **do**
3:       Calculate pseudo-labels $\hat{y}$ using Eq. (4) and (5)
4:       Calculate domain labels $\hat{d}$ using Eq. (7)
5:       Update $\Theta_{stu}$ of $\mathcal{E}_{stu}(\cdot)$ and $\Phi_{stu}(\cdot)$ using Eq. (11)
6:       Update $\Theta_{tea}$ of $\mathcal{E}_{tea}(\cdot)$ and $\Phi_{tea}(\cdot)$ using Eq. (9)
7:    **end for**
8: **end for**
9: Modify pseudo-labels $\hat{y}$ into $y_{mpl}$ using Eq. (12)
10: Calculate ranking scores $y_{rank}$ using Eq. (15)
11: **for** each epoch **do**
12:    **for** each batch in $\mathcal{S}_T$ **do**
13:       Calculate predictions $\hat{y}_{ret}$ using Eq. (13)
14:       Update parameters of Ret$(\cdot)$ using Eq. (17)
15:    **end for**
16: **end for**

---

while maximizing the loss for domain classification ($\mathcal{L}_{dom}$). The total loss is defined as follows:

$$\mathcal{L}_{total} = \mathcal{L}_{ce} + \lambda_1 * \mathcal{L}_{sup} - \lambda_2 * \mathcal{L}_{dom}, \quad (11)$$

where $\lambda_1$ and $\lambda_2$ are the weights of $\mathcal{L}_{sup}$ and $\mathcal{L}_{dom}$, respectively.

### 4.3 Hybrid Training

In this module, we use a hybrid training mechanism to obtain the retrieval model Ret$(\cdot)$, as depicted in Algorithm 1. Following the previous steps, samples in the target domain $\mathcal{S}_{sam}$ are assigned pseudo-labels $\hat{y}^t$, while samples in the source domain have both pseudo-labels $\hat{y}^s$ and ground-truth labels $y^s$. To minimize the noise in the retrieval model training and emphasize the importance of positive samples, we continue to use $y^s = 1$ as the training labels for positive exposed samples. For other samples, we utilize pseudo-labels as the targets. Thus, for the training sample set $\mathcal{S}_T = \mathcal{S}_{exp} \cup \mathcal{S}_{sam}$, the modified pseudo-labels $y_{mpl}$ is defined as follows:

$$y_{mpl} = \begin{cases} 1 & \text{if } (u^s, i^s, y^s, e = 1) \in \mathcal{S}_{exp} \ \& \ y^s = 1, \\ \hat{y} & \text{others.} \end{cases} \quad (12)$$

Then, we input each training sample $(u, i, y_{mpl}) \in \mathcal{S}_T$ into the retrieval model Ret$(\cdot)$ to obtain the predictions:

$$\hat{y}_{ret} = \text{Ret}(u, i). \quad (13)$$

For the retrieval model training, we measure the difference between the predictions $\hat{y}_{ret}$ and the modified pseudo-labels $y_{mpl}$ as follows:

$$\mathcal{L}_{mpl} = -\frac{1}{|\mathcal{S}_T|} \sum_{\mathcal{S}_T} y_{mpl} \log \hat{y}_{ret} + (1 - y_{mpl}) \log(1 - \hat{y}_{ret}), \quad (14)$$

where $|\mathcal{S}_T|$ is the number of training samples. Additionally, we input each training sample into the pre-trained model Rank$(\cdot)$ to obtain the ranking scores:

$$y_{rank} = \text{Rank}(u, i). \quad (15)$$

**Table 1: Statistics of datasets with multi-modal features, i.e., Visual (V), Acoustic (A), and Textual (T) information.**

| Dataset | #Users | #Items | #Interactions | Modality |
|---------|--------|--------|---------------|----------|
| WeChat  | 16,904 | 42,031 | 1,639,527     | V, A, T  |
| TikTok  | 38,883 | 49,842 | 1,502,827     | V, A, T  |

We employ a knowledge distillation strategy to narrow the gap between the output distributions of the retrieval model and the ranking model [52], ensuring consistency in the cascade system:

$$\mathcal{L}_{con} = \frac{1}{|\mathcal{S}_T|} \sum_{\mathcal{S}_T} (y_{rank} - \hat{y}_{ret})^2. \quad (16)$$

Finally, our minimization objective is defined as follows:

$$\mathcal{L}_{train} = \mathcal{L}_{mpl} + \lambda_3 * \mathcal{L}_{con}, \quad (17)$$

where $\lambda_3$ is the weight of $\mathcal{L}_{con}$.

## 5 EXPERIMENTS

To deeply evaluate DAMCAR, we conduct extensive experiments with the aim of answering the following research questions:

- RQ1: How does DAMCAR perform compared with baselines?
- RQ2: What is the role of each designed module in DAMCAR?
- RQ3: What is the impact of hyperparameters on performance?
- RQ4: Why does DAMCAR actually work in mitigating SSB?
- RQ5: How effective is DAMCAR in real-world applications?

### 5.1 Experimental Setup

*5.1.1 Datasets.* We conduct a series of experiments on two video recommendation datasets with the required multi-modal features, i.e., WeChat[2] and TikTok[3]. Table 1 summarizes the statistics.

- **WeChat**: This dataset is collected from the short videos platform of WeChat[4], recording the user behaviors in two weeks. It contains pre-trained multi-modal feature embeddings with visual, acoustic, and textual information.
- **TikTok**: This dataset contains user interactions with short videos, collected from the TikTok[5] platform. The multi-modal features are the visual, acoustic, and textual content of videos.

Following the settings of previous works [46, 47, 56], we divide the datasets into training, validation, and test sets according to the timestamps of user interactions in a ratio of 6:2:2. Moreover, we set the ground-truth label to finish playing ($y = 1$) or not ($y = 0$).

*5.1.2 Testbed.* We build a two-stage cascade recommendation system as the testbed for performance comparison [48, 50], consisting of retrieval and ranking stages. In the retrieval stage, we utilize Deep Structured Semantic Model (DSSM) [28] as the backbone, a dual-tower structural model extensively applied in industrial scenarios. The dimensions of the user feature embedding and the item feature embedding are both set to 128. In the ranking stage, we employ two common CTR models, i.e., Deep Factorization-Machine

---

[2]https://algo.weixin.qq.com/2021/problem-description
[3]https://www.biendata.com/competition/icmechallenge2019
[4]https://www.wechat.com/en
[5]https://www.tiktok.com

**Table 2: Performance comparison of different methods in a two-stage cascade recommendation system, where DSSM is used for retrieval and DeepFM is used for ranking. The best and second-best results are marked in bold and underlined, respectively.**

| Stage | Method | WeChat | | | | | | TikTok | | | | | |
|---|---|---|---|---|---|---|---|---|---|---|---|---|---|
| | - | R@100 | P@100 | F@100 | R@200 | P@200 | F@200 | R@100 | P@100 | F@100 | R@200 | P@200 | F@200 |
| Ret. | BC | 0.0740 | 0.0113 | 0.0196 | 0.1375 | 0.0106 | 0.0197 | 0.0627 | 0.0063 | 0.0114 | 0.1222 | 0.0062 | 0.0118 |
| | KD | 0.0785 | 0.0121 | 0.0210 | 0.1386 | 0.0110 | 0.0204 | 0.0664 | 0.0066 | 0.0120 | 0.1347 | 0.0068 | 0.0129 |
| | TL | 0.0855 | 0.0125 | 0.0218 | 0.1449 | 0.0110 | 0.0204 | 0.0645 | 0.0065 | 0.0118 | 0.1409 | 0.0070 | 0.0133 |
| | AR | 0.0823 | 0.0126 | 0.0219 | 0.1511 | 0.0116 | 0.0215 | 0.0664 | 0.0066 | 0.0120 | 0.1523 | 0.0074 | 0.0141 |
| | MUDA | 0.0812 | 0.0126 | 0.0218 | 0.1433 | 0.0113 | 0.0209 | 0.0685 | 0.0071 | 0.0129 | 0.1540 | 0.0075 | 0.0143 |
| | DAMCAR | **0.0884** | **0.0135** | **0.0234** | **0.1579** | **0.0121** | **0.0225** | **0.0775** | **0.0075** | **0.0137** | **0.1580** | **0.0077** | **0.0147** |
| | - | N@10 | M@10 | H@10 | N@20 | M@20 | H@20 | N@10 | M@10 | H@10 | N@20 | M@20 | H@20 |
| Rank. | BC | 0.0477 | 0.0284 | 0.0124 | 0.0718 | 0.0336 | 0.0130 | 0.0356 | 0.0274 | 0.0066 | 0.0477 | 0.0300 | 0.0062 |
| | KD | 0.0549 | 0.0329 | 0.0142 | 0.0763 | 0.0364 | 0.0146 | 0.0402 | 0.0317 | 0.0072 | 0.0526 | 0.0341 | 0.0067 |
| | TL | 0.0507 | 0.0288 | 0.0138 | 0.0652 | 0.0296 | 0.0121 | 0.0402 | 0.0298 | 0.0078 | 0.0512 | 0.0323 | 0.0066 |
| | AR | 0.0562 | 0.0340 | 0.0148 | 0.0782 | 0.0386 | 0.0142 | 0.0400 | 0.0313 | 0.0073 | 0.0516 | 0.0338 | 0.0064 |
| | MUDA | 0.0545 | 0.0327 | 0.0147 | 0.0746 | 0.0338 | 0.0143 | 0.0413 | 0.0323 | 0.0076 | 0.0543 | 0.0350 | 0.0068 |
| | DAMCAR | **0.0657** | **0.0394** | **0.0166** | **0.0867** | **0.0441** | **0.0154** | **0.0443** | **0.0356** | **0.0089** | **0.0558** | **0.0359** | **0.0078** |

**Table 3: Performance comparison of different methods in a two-stage cascade recommendation system, where DSSM is used for retrieval and DCN is used for ranking. The best and second-best results are marked in bold and underlined, respectively.**

| Stage | Method | WeChat | | | | | | TikTok | | | | | |
|---|---|---|---|---|---|---|---|---|---|---|---|---|---|
| | - | R@100 | P@100 | F@100 | R@200 | P@200 | F@200 | R@100 | P@100 | F@100 | R@200 | P@200 | F@200 |
| Ret. | BC | 0.0740 | 0.0113 | 0.0196 | 0.1375 | 0.0106 | 0.0197 | 0.0627 | 0.0063 | 0.0114 | 0.1222 | 0.0062 | 0.0118 |
| | KD | 0.0825 | 0.0127 | 0.0220 | 0.1467 | 0.0114 | 0.0212 | 0.0655 | 0.0065 | 0.0118 | 0.1284 | 0.0065 | 0.0124 |
| | TL | 0.0793 | 0.0124 | 0.0214 | 0.1458 | 0.0114 | 0.0211 | 0.0670 | 0.0066 | 0.0120 | 0.1415 | 0.0070 | 0.0133 |
| | AR | 0.0821 | 0.0124 | 0.0215 | 0.1526 | 0.0117 | 0.0217 | 0.0709 | 0.0069 | 0.0126 | 0.1445 | 0.0071 | 0.0135 |
| | MUDA | 0.0834 | 0.0132 | 0.0228 | 0.1458 | 0.0118 | 0.0218 | 0.0704 | 0.0072 | 0.0131 | 0.1566 | 0.0075 | 0.0143 |
| | DAMCAR | **0.0917** | **0.0135** | **0.0235** | **0.1620** | **0.0124** | **0.0230** | **0.0768** | **0.0076** | **0.0138** | **0.1579** | **0.0078** | **0.0149** |
| | - | N@10 | M@10 | H@10 | N@20 | M@20 | H@20 | N@10 | M@10 | H@10 | N@20 | M@20 | H@20 |
| Rank. | BC | 0.0494 | 0.0294 | 0.0129 | 0.0732 | 0.0335 | 0.0137 | 0.0374 | 0.0303 | 0.0064 | 0.0474 | 0.0319 | 0.0057 |
| | KD | 0.0606 | 0.0389 | 0.0150 | 0.0761 | 0.0386 | 0.0137 | 0.0388 | 0.0315 | 0.0066 | 0.0510 | 0.0329 | 0.0066 |
| | TL | 0.0577 | 0.0358 | 0.0145 | 0.0733 | 0.0370 | 0.0133 | 0.0375 | 0.0289 | 0.0069 | 0.0509 | 0.0310 | 0.0068 |
| | AR | 0.0596 | 0.0382 | 0.0149 | 0.0790 | 0.0399 | 0.0146 | 0.0429 | 0.0333 | 0.0079 | 0.0544 | 0.0343 | 0.0071 |
| | MUDA | 0.0639 | 0.0411 | 0.0162 | 0.0729 | 0.0346 | 0.0139 | 0.0406 | 0.0307 | 0.0078 | 0.0511 | 0.0318 | 0.0068 |
| | DAMCAR | **0.0706** | **0.0455** | **0.0177** | **0.0824** | **0.0408** | **0.0154** | **0.0465** | **0.0374** | **0.0092** | **0.0574** | **0.0371** | **0.0080** |

(DeepFM) [22] and Deep & Cross Network (DCN) [55], respectively, achieving precise ranking by deeply capturing feature interactions.

5.1.3  **Baselines**. Referring to [48, 56], we compare DAMCAR with five baselines: (i) **Binary Classification (BC)** is considered as the base method for comparison, training the retrieval model on exposed data, which does not account for SSB. (ii) **Knowledge Distillation (KD)** uses the ranking model's predictions as the ground-truth labels for retrieval model training [48]. (iii) **Transfer Learning (TL)** utilizes unexposed data with the pseudo-labels generated by the ranking model to fine-tune the item's embedding tower and keep the user tower unchanged [42]. (iv) **Adversarial Regularization (AR)** trains a domain classifier from the intermediate output of the retrieval model and uses loss's negative as regularization [21]. (v) **Modified Unsupervised Domain Adaptation (MUDA)** uses unexposed data with the pseudo-labels derived by transforming the ranking model's predictions into binary classes for retrieval model training [56].

5.1.4  **Evaluation Metrics**. In the retrieval stage, we adopt three widely used metrics, i.e., Recall (R@K), Precision (P@K), and F1@K (F@K), which measure the quality of relevant candidates returned by the retrieval model. In the ranking stage, Normalized Discounted Cumulative Gain (N@K), Mean Average Precision (M@K), and Hit Ratio (H@K) are utilized to assess the ranking performance. Detailed calculations for these metrics can be found in [39]. To simulate different recommendation scenarios, we set K to 100/200 for retrieval and 10/20 for ranking, respectively.

5.1.5  **Implement Details**. For all methods, we use AdaGrad optimizer [14] with an initial learning rate of $1e^{-2}$. The batch size is 512 for training and 128 for testing. In DAMCAR, we set $R = 25$ and $H = 50$ on WeChat, while $R = 30$ and $H = 60$ on TikTok, for the target domain generation. The encoder $\mathcal{E}(\cdot)$ consists of a three-layer MLP with hidden dimensions of 300, 200, and 100, while the linear transformation $g(\cdot)$ utilizes a two-layer MLP with hidden dimensions of 128 and 100, both using ReLU [20] as the activation function. For

**Table 4: Loss weights tuned in the experiments.**

| Variable | Value Range | Description |
|---|---|---|
| $\lambda_1$ | $\{5e^{-1}, 6e^{-1}, 7e^{-1}, 8e^{-1}, 9e^{-1}\}$ | Weight of $\mathcal{L}_{sup}$ |
| $\lambda_2$ | $\{1e^{-1}, 2e^{-1}, 3e^{-1}, 4e^{-1}, 5e^{-1}\}$ | Weight of $\mathcal{L}_{dom}$ |
| $\lambda_3$ | $\{1e^{-1}, 2e^{-1}, 3e^{-1}, 4e^{-1}, 5e^{-1}\}$ | Weight of $\mathcal{L}_{con}$ |

the label generator $\Phi(\cdot)$ and the domain classifier $\Psi(\cdot)$, we use a
single-layer MLP with the Sigmoid [23] activation function. We use
GRL [18] to reverse gradients for matching the update direction.
The smoothing parameter $\alpha$ is set to 0.99. Furthermore, we use grid
search to find the best weights of the three different losses. The
value ranges are shown in Table 4.

## 5.2 Performance Comparison (RQ1)

Table 2 and Table 3 present the results of DAMCAR and baselines
in two different cascade recommendation systems, respectively. In
Table 2, DSSM is employed in the retrieval stage, while DeepFM is
used in the ranking stage. To evaluate the generalization capability
of our approach, Table 3 maintains the same retrieval model but
substitutes the ranking model with DCN.

From the experimental results, we have the following observa-
tions: (i) Compared with the base method BC, DAMCAR gains
significant improvements in both retrieval and ranking metrics.
That is, it correctly retrieves a sufficiently large number of candi-
dates and precisely ranks them to be delivered to users, implying
that mitigating SSB can effectively enhance recommendation per-
formance. (ii) DAMCAR outperforms all debiasing baselines on
two datasets and under different settings of K. We attribute the
performance gain to three factors. Firstly, DAMCAR collects a high-
quality subset from unexposed data as the target domain, for data
augmentation. Compared with existing sampling methods, our tar-
get domain samples contain richer information in enhancing the
retrieval model training. Secondly, DAMCAR employs an effective
mechanism to generate robust pseudo-labels, in contrast to rely-
ing solely on the ranking model's predictions, e.g., TL and MUDA.
The performance improvement of AR over BC further supports
the necessity of adversarial learning. Thirdly, the results of KD
show that ensuring consistency in the cascade system benefits the
final recommendation performance, and DAMCAR takes this as
one of the training objectives through a hybrid mechanism. (iii) De-
spite employing different ranking models, DAMCAR enhances the
overall recommendation performance under both configurations.
This improvement indicates that enhanced retrieval performance
positively impacts the ranking model's ability.

## 5.3 Ablation Study (RQ2)

In this part, we conduct the ablation study to explore the contribu-
tion of each designed module in DAMCAR. We compare DAMCAR
with three variants: (i) w/o TDG, where target domain generation is
replaced by randomly sampling the same number of samples from
unexposed data; (ii) w/o EMA, i.e., without using EMA to create a
teacher model that guides the generation of pseudo-labels for target
domain samples; and (iii) w/o MPL, which only uses the generated

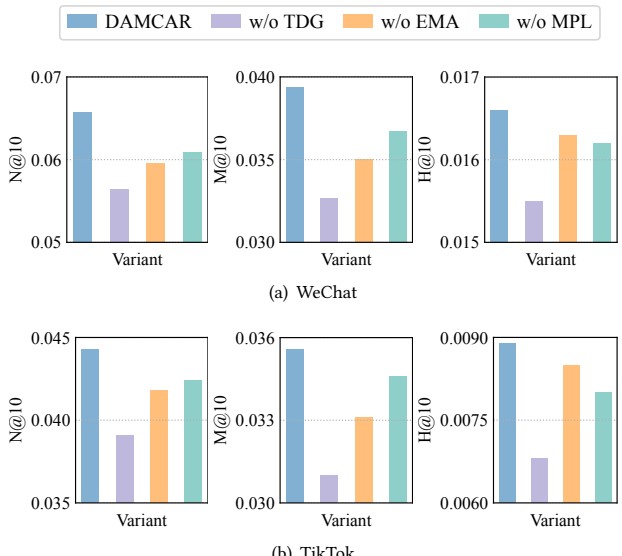

(a) WeChat

(b) TikTok

**Figure 3: Results of the ablation study on WeChat and TikTok,
using DSSM and DeepFM as the backbone models.**

pseudo-labels without modifying them by ground-truth labels. Fig-
ure 3 shows their final ranking performance on two datasets, using
DSSM and DeepFM as the backbone models.

These results illustrate two facts: (i) All designed modules are
important to DAMCAR. The ranking performance will deteriorate if
any module is removed. For example, in Figure 3(a) and Figure 3(b),
each variant has worse performance in terms of N@10, M@10,
and H@10 on WeChat and TikTok, respectively. (ii) Among the
three modules, target domain generation has the greatest impact
on performance, because generating a high-quality target domain
is an important basis for generating pseudo-labels and performing
hybrid training. Compared with random sampling, which often fails
to select samples close to the decision boundary, our method se-
lects samples that significantly enhance the model's discriminative
ability. In addition, incorporating an EMA-based supervision mech-
anism and enhancing the model's perception of positive exposed
samples through MPL can further improve the performance.

## 5.4 Hyperparameters Analysis (RQ3)

In Figure 4, we investigate the impact of several key hyperparame-
ters on performance for DAMCAR.

*5.4.1* ***Effects of R and H.*** We first evaluate the ranking quality
of DAMCAR under different values of the radius $R$ and the number
of visits $H$, i.e., two crucial settings for target domain generation.
From the results, we can find that the metrics first rise and then fall,
with the optimal settings of $R = 25$ and $H = 50$ on WeChat, while $R$
= 30 and $H = 60$ on TikTok. We attribute the variation of settings
to differences in dataset sparsity. For the hyperparameters $R$ and $H$,
they determine the search range and number of the random walk
algorithm. Excessively large values may gather samples far from the
model's decision boundary, while overly small values fail to collect
adequate informative samples, both of which hinder the generation

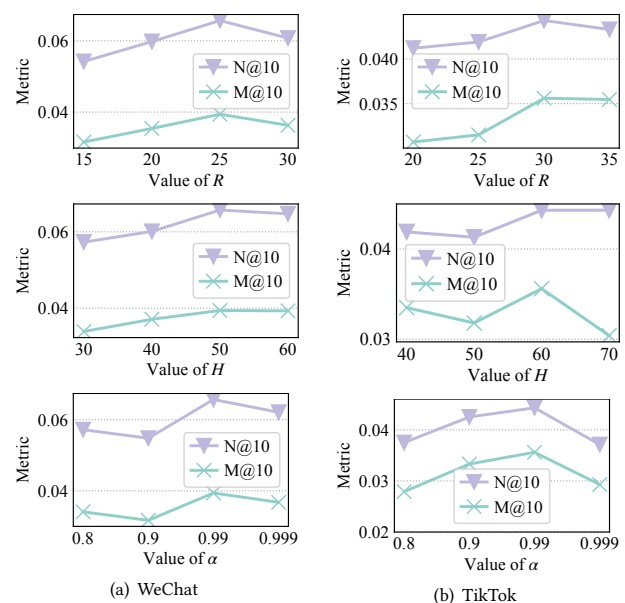

**Figure 4: Performance comparison under different hyperparameter settings on WeChat and TikTok.**

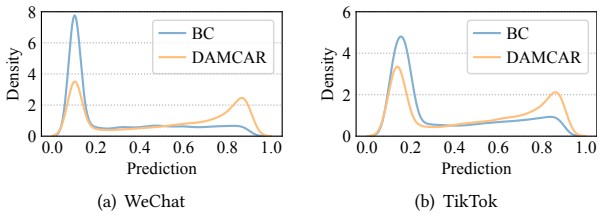

(a) WeChat

(b) TikTok

**Figure 5: Visualization of predictions for the unexposed samples. BC tends to regard them as negative, while DAMCAR mines potential positive samples.**

of a high-quality target domain, leaving the hard-to-distinguish unexposed samples without correct attention.

*5.4.2* **Effects of** $\alpha$. Additionally, we investigate DAMCAR's performance under different values of the smoothing parameter $\alpha$. $\alpha$ is used to control the update rate of the teacher model created by EMA. Results indicate that $\alpha = 0.99$ yields superior performance. Larger values may impede effective model updates, while smaller values compromise model robustness.

## 5.5 Visualization and Further Analysis (RQ4)

To further verify why DAMCAR works in mitigating SSB, we show the prediction distribution of the retrieval model. In Figure 5, we train the retrieval model on WeChat and TikTok using both DAMCAR and BC, then showcase the predictions for target domain samples. We can find that BC tends to recognize unexposed data as negative samples, leading to inconsistency between training and inference. In contrast, DAMCAR effectively uncovers potential positive samples (i.e., unexposed positive samples) and delivers them to

**Table 5: Average improvement of DAMCAR in online A/B testing. We deploy our approach in the retrieval model of a multimedia video recommendation system.**

| Metric | Confidence Interval | Improvement |
|---|---|---|
| Watch Time | [0.03%, 0.25%] | +0.14% |
| Like | [0.06%, 0.84%] | +0.45% |
| Follow | [0.12%, 0.41%] | +0.27% |
| Share | [0.14%, 0.49%] | +0.31% |
| Vertical Category | [0.07%, 1.04%] | +0.55% |

users. DACMAR rectifies the distribution discrepancy between the training and inference data, thus achieving unbiased personalized recommendations. Unlike indiscriminate labeling of unexposed samples as negative, DAMCAR harnesses sufficient information in unexposed data to enhance the retrieval model training.

## 5.6 Results of the Online Deployment (RQ5)

To validate the effectiveness of our approach in real-world applications, we deploy it on the cascade recommendation system for an online multimedia video platform and conduct A/B testing. Since each module of DAMCAR is implemented independently, it does not affect the training efficiency of the retrieval model. Specifically, we generate the target domain from historical data and train a separate DNN model to produce corresponding pseudo-labels. These offline-generated data and corresponding labels are then utilized to enhance the retrieval model for mitigating SSB.

We provide metrics in three dimensions to comprehensively evaluate the performance of DAMCAR: (i) Engagement: Measure user engagement with the platform. We use Watch Time as a metric to quantify the total time spent by users. (ii) Interaction: Measure user satisfaction with the recommended results. We employ three common metrics, i.e., Like (clicking the like button), Follow (following the video creator), and Share (sharing the video with others). (iii) Diversity: Measure the variety of content consumed by users. We use Vertical Categories to signify the diversity of presented content, representing the number of unique video categories shown to all users. The results of online A/B testing are shown in Table 5, demonstrating the effectiveness of our approach in industrial scenarios for an improved user experience.

## 6 CONCLUSION

In this paper, we introduce a debiasing framework named DAMCAR, as a comprehensive solution to mitigate Sample Selection Bias (SSB) in multimedia cascade recommendation systems. We start by sampling a target domain from unexposed data and use adversarial domain adaptation to generate unbiased pseudo-labels for target domain samples, narrowing the gap between training and inference data distributions. To further enhance the robustness of models and the reliability of pseudo-labels, we employ Exponential Moving Average (EMA) to create a teacher model, supervising the learning of pseudo-labels through a self-distillation mechanism. Experiments conducted on two real-world datasets and the online deployment of an industrial multimedia cascade recommendation system prove the practical benefits of DAMCAR, significantly improving recommendation effectiveness without sacrificing efficiency.

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
