# OpenReview forum: "Mitigating Sample Selection Bias with Robust Domain Adaption in Multimedia Recommendation"
_acmmm.org/ACMMM/2024/Conference — MM2024 Oral_

### Official Review · Reviewer_D9Nx · 2024-05-16

**Rating:** 5
**Confidence:** 3

**Summary:**

This paper presents a new method to solve the problem of sample selection bias in multimedia cascade recommendation. First, the author performs a random walk algorithm in the user-item bipartite graph to determine the hard-to-distinguish samples other than the exposed samples. Secondly, the author uses adversarial generation method to generate pseudo-labels for the above samples, and iterates the model parameters according to batches to improve the quality of pseudo-labels. Finally, the authors trained all the samples using a hybrid training strategy. Comparison experiments, ablation experiments and parametric experiments were performed, and online A/B tests were used to demonstrate the validity of the model.

**Strengths:**

1. The method of using adversarial generation to reduce sample selection bias is feasible and reflects its innovation. Specifically, the paper uses a domain classifier to try to classify exposed and unexposed samples, and pits the encoder against the domain classifier. This is a more natural way of distributing false labels, in line with the intuition that exposed and unexposed samples converge to the same distribution.
2. This article carries out extensive experiments. In addition to comparison experiment, parameter experiment and ablation experiment, the author carried out visual analysis of the model effect. At the same time, the author conducts A/B test of online deployment, which fully proves that the model has good performance in practical application.
3. The article is well organized and the language is clear. The author is able to clearly demonstrate the challenges, innovations and specific steps of the algorithm through formal language.

**Limitations:**

1. For the random walk algorithm of the bipartite graph, it is suggested to give further explanation. This algorithm is used in the section 'Target Domain Generation' to extract hard-to-distinguish samples. Maybe there are other algorithms that perform the same function. The paper can illustrate the advantages of this algorithm, even if the ablation experiment proves it to a certain extent.
2. An adversarial training method is used to generate false labels, and the parameters are adjusted through multi-stage training, which may have a relatively large impact on the calculation cost. I wonder the machine parameters of the experiment and the difference in the algorithm's running time compared to other benchmark methods.
3. Beyond the discussion of this paper, there are several papers that are relevant and important. Authors should cite these articles and discuss them in an 'introduction' or 'Related work' section. Such as the following:
- Mitigating Confounding Bias in Practical Recommender Systems With Partially Inaccessible Exposure Status. 2023.
- Be causal: De-biasing social network confounding in recommendation. 2023.
- Implicit feedbacks are not always favorable: Iterative relabeled one-class collaborative filtering against noisy interactions. 2021.
4. Besides, authors should not use a bracketed year when citing literature. This problem arises in references [5], [7], [12], [14], [21], [22], [24], [25], [30], [35], and[41].

**Suitability:**

2

---

### Official Review · Reviewer_asDf · 2024-05-21

**Rating:** 4
**Confidence:** 4

**Summary:**

The paper proposes a debiasing framework called DAMCAR to mitigate sample selection bias in multimedia recommendation systems that utilize cascade architectures with retrieval and ranking stages. Sample selection bias arises due to the distribution discrepancy between the exposed training data used for the retrieval model and the mostly unexposed candidates it needs to score during inference. DAMCAR bridges this gap through domain adaptation techniques. It has three modules: 1) Target domain generation, where it constructs a weighted user-item bipartite graph and uses random walks to sample hard-to-distinguish unexposed data as an informative target domain. 2) Robust pseudo-label generation, employing adversarial domain adaptation to align feature representations of the source (exposed) and target (sampled unexposed) domains, and using exponential moving average to create a teacher model that enhances pseudo-label quality via self-distillation. 3) Hybrid training that combines the generated pseudo-labels with scores from the ranking model to train the retrieval model robustly for stable inference performance. Comprehensive experiments on datasets and an online recommendation system demonstrate DAMCAR's effectiveness.

**Strengths:**

1. The idea of applying unsupervised domain adaptation techniques to mitigate sample selection bias in multimedia recommendation systems is a novel contribution. Previous debiasing methods either relied on negative sampling/treating unexposed as negative which can be noisy, or required expensive collection of unbiased datasets.
2. Leveraging graph connectivity to identify hard-to-distinguish unexposed samples for the target domain is an interesting perspective that exploits multi-hop relationships.
3. The technical details like constructing the bipartite user-item graph, transition probabilities, random walk sampling algorithm etc. seem pragmatic.
4. The paper conducts offline experiments on two real-world datasets and implements DAMCAR in a practical multimedia video recommendation system and reports positive results from online A/B testing.

**Limitations:**

1. There are some design choices made in DAMCAR's modules that could benefit from deeper analysis, such as, the specific random walk sampling algorithm used for target domain generation, the purpose of designing g() in Equation 5.
2. All datasets used for validation are not publicly available (the datasets are not available at the URLs in the paper), so a more detailed description of the datasets is needed to demonstrate the general applicability of the method.

**Suitability:**

3

---

### Official Review · Reviewer_wpH3 · 2024-05-24

**Rating:** 4
**Confidence:** 2

**Summary:**

This paper introduces DAMCAR, a debiasing framework designed to mitigate Sample Selection Bias (SSB) in multimedia cascade recommendation systems by leveraging domain adaptation. The framework comprises three primary modules: target domain generation, robust pseudo-label generation, and hybrid training. By sampling a target domain from unexposed data and employing adversarial domain adaptation, DAMCAR generates unbiased pseudo-labels for target domain samples, effectively narrowing the gap between training and inference data distributions. The framework also uses Exponential Moving Average (EMA) to create a teacher model that enhances the robustness of models and the reliability of pseudo-labels. Extensive experiments on real-world datasets and an online deployment demonstrate that DAMCAR significantly improves recommendation effectiveness without sacrificing efficiency, offering a practical solution for industrial applications facing issues with SSB.

**Strengths:**

1. The research direction presented in the paper is both meaningful and holds significant potential for generating positive impacts.
2. The experimental results indicate that the proposed method consistently outperforms the baseline approaches, highlighting its overall superiority.
3. The paper is well-organized and clearly written.

**Limitations:**

1. Not provide a thorough exploration or validation of the random walk algorithm used in target domain generation, particularly concerning its effectiveness in accurately identifying hard-to-distinguish samples near the decision boundary.
2. The paper tests its proposed framework, DAMCAR, exclusively on DSSM, DeepFM, and DCN models, which are not the most current state-of-the-art models in the field of recommendation systems. Including experiments with more advanced and recent recommendation models could strengthen the evidence of the framework's generalizability and effectiveness across different architectures.

**Suitability:**

3

---

### Meta-Review · Area_Chair_uuze · 2024-07-02

**Recommendation:** Accept (Oral)
**Confidence:** 5

**Metareview:**

Addressing the Sample Selection Bias in a multimedia cascade recommendation system is an important task. In this paper, the authors leverage the domain adaptation. The experimental results indicate that the proposed method consistently outperforms the baseline approaches. Overall, the paper is well-organized and well-written.